# Towards Machine Learning in Heterogeneous Catalysis—A Case Study of 2,4-Dinitrotoluene Hydrogenation

**DOI:** 10.3390/ijms241411461

**Published:** 2023-07-14

**Authors:** Alexandra Jakab-Nácsa, Attila Garami, Béla Fiser, László Farkas, Béla Viskolcz

**Affiliations:** 1BorsodChem Ltd., Bolyai tér 1, H-3700 Kazincbarcika, Hungary; alexandra.nacsa@borsodchem.eu (A.J.-N.); laszlo.farkas@borsodchem.eu (L.F.); 2Institute of Chemistry, Faculty of Materials Science and Engineering, University of Miskolc, H-3515 Miskolc-Egyetemváros, Hungary; 3Institute of Energy, Ceramics and Polymer Technology, University of Miskolc, H-3515 Miskolc, Hungary; attila.garami@uni-miskolc.hu; 4Higher Education and Industrial Cooperation Centre, University of Miskolc, H-3515 Miskolc, Hungary; bela.fiser@uni-miskolc.hu; 5Ferenc Rakoczi II Transcarpathian Hungarian College of Higher Education, 90200 Beregszász, Transcarpathia, Ukraine; 6Department of Physical Chemistry, Faculty of Chemistry, University of Lodz, 90-236 Lodz, Poland

**Keywords:** machine learning, exploratory data analysis, catalyst ranking, catalyst design, MIRA21

## Abstract

Utilization of multivariate data analysis in catalysis research has extraordinary importance. The aim of the MIRA21 (MIskolc RAnking 21) model is to characterize heterogeneous catalysts with bias-free quantifiable data from 15 different variables to standardize catalyst characterization and provide an easy tool to compare, rank, and classify catalysts. The present work introduces and mathematically validates the MIRA21 model by identifying fundamentals affecting catalyst comparison and provides support for catalyst design. Literature data of 2,4-dinitrotoluene hydrogenation catalysts for toluene diamine synthesis were analyzed by using the descriptor system of MIRA21. In this study, exploratory data analysis (EDA) has been used to understand the relationships between individual variables such as catalyst performance, reaction conditions, catalyst compositions, and sustainable parameters. The results will be applicable in catalyst design, and using machine learning tools will also be possible.

## 1. Introduction

One hundred years ago, catalysts were considered a rarity, but today they play pivotal role in modern industry. The global catalyst market reached about USD 34 billion in 2020 [1]. Sustainability and green chemistry efforts, through renewable energy resources and environmentally friendly materials, pose new challenges to catalyst developers in R&D sectors [2]. There is no shortage of catalytic processes in isocyanate production either [3]. One of the main technological processes of isocyanate production is the catalytic hydrogenation of aromatic nitro compounds into amine compounds such as aniline and toluene diamine.

More than 7000 scientific articles have been published with the terms “nitro, aromatic, hydrogenation, catalyst” in only the last 5 years [4]. Several heterogeneous catalysts with different carrier compositions have already been developed for nitroaromatic hydrogenation, starting with activated carbon- or carbon black-supported catalysts and extending through various metal-oxide-supported and transition-metal-oxide-supported systems, including zeolite- and polymer-based catalysts [5,6,7,8,9,10]. The active compounds and their combinations applied to the catalyst support also vary widely [11,12,13,14,15]. 

Navigating catalytic research results at the beginning of a catalyst development process is a great challenge. More and more diversified combinations of heterogeneous catalysts have been prepared to achieve maximum efficiency. Thus, more and more information is available, but it is not obvious how we can use this large number of data properly. In most cases, review articles have been created about a catalytic reaction in which the results have been organized based on a few selected properties [16,17,18]. Catalytic data have been presented differently in the different articles. Therefore, it is challenging to compare different type of catalyst characterization data, hydrogenation test results in various applied reaction conditions, and catalytic performance properties with sometimes divergent interpretations. 

To overcome such difficulties, our first goal was to create a catalyst library that not only stores data but provides information in an easily comparable manner. The MIRA21 (MIskolc RAnking 21) model has been established for the characterization, comparison, and ranking of catalysts with a mathematical model that is based on a descriptor system [19]. In our previous work, hydrogenation reactions of nitrobenzene and 2,4-dinitrotoluene were examined based on four groups of parameters such as catalyst performance, reaction conditions, catalyst conditions, and sustainability parameters [20]. The setting of the parameter set was compiled based on the information available in various scientific publications. Since the independence of the parameters was not examined previously, the aim of this research was to carry out statistical analysis using the generated database. 

We are living in the golden age of artificial intelligence in various scientific fields, including chemistry [21,22]. Nowadays, solving multivariate data analysis problems with machine learning methods—within AI—is becoming more and more common among researchers [23,24,25]. Exploratory data analysis (EDA) is the application of several statistical techniques aimed at investigating, describing, and summarizing the nature of data. This allows us to identify possible errors, reveal the existence of an outlier, check the relationship between variables (correlations) and their possible redundancy, and conduct a descriptive analysis of data using graphical representations and summaries of the most important aspects [26]. EDA of previous catalytic data reveals the exploration of correlations between the physicochemical properties and performance of catalysts [27,28,29,30,31,32,33,34]. Through this research, we would like to show how to reach conscious data analysis starting from data collection guided by chemical intuition and achieving results that can be applied to construct machine learning algorithms. 

Exploratory data analysis of the literature data of 2,4-dinitrotoluene hydrogenation catalysts was carried out. Based on the MIRA21 model, data was extracted from scientific articles and converted into a catalyst library [20]. Descriptive and statistical analysis provides an opportunity to study the MIRA21 model descriptor system and the correlations between catalyst physicochemical properties and catalyst performance parameters such as conversion and selectivity. Furthermore, a workflow was created which can be used to carry out the EDA of available literature data of other heterogeneous catalysts. 

## 2. Results and Discussion

### 2.1. Future Selection Analysis

First, the relationship between the different variables was examined, and then patterns were searched in the system regarding the composition of the catalyst. As a result of the correlation examination of properties describing the catalyst, information was collected for the ranking. By mapping the patterns, the composition of the catalysts that can provide excellent performance during the hydrogenation of 2,4-dinitrotoluene can be predicted. 

Since there is a relatively low number of data available for this reaction, the results carry some uncertainty. However, the applicability of the exploratory data analysis procedure to other reactions is undeniable. 

### 2.2. Data Distributions

Prior to the correlation analysis, the applied numerical dataset has to be examined. A violin plot of the catalyst library was created by using the studied parameters (Figure 1). This special type of plot can show the median of data, the interquartile ranges, and the entire distribution of different data values. The width of the curves means the frequency of data points. 

The sample was heterogeneous with respect to data values. Based on the plot diagrams of the various parameters, it can be concluded that the data do not follow a normal distribution. In most cases, the shape of the violin is much more similar to the bimodal distribution or the log-normal distribution with its asymmetry. There is also a tapering part of the shape in the case of maximum conversion (*X*), time (*t*), and the amount of catalyst (CAT), which indicates that there may be outliers in the system. It is interesting to note that two larger peaks were observed within the reaction temperature data with a quite big difference.

In order to ensure that outliers did not affect the correlation analysis, the data were filtered. By plotting the data, outliers were selected and removed from the dataset. This filtering was performed by defining the limit values for justified parameters. Catalysts with a maximum conversion greater than 50%, yield greater than 50%, reaction time shorter than 240 min, and pressure less than 50 atm were analyzed. Unfortunately, the data cleaning process reduces the number of available data, but it also reduces the distortion of the correlation analysis results.

### 2.3. Correlation Analysis 

In the correlation analysis, the first step is to make sure that the following five assumptions are met for calculating the Pearson coefficient [35]: Level of Measurement: The two variables should be measured at the interval or ratio level.Linear Relationship: There should exist a linear relationship between the two variables.Normality: Both variables should be roughly normally distributed.Related Pairs: Each observation in the dataset should have a pair of values.No Outliers: There should be no extreme outliers in the dataset [36].

Based on the violin plot, the normality is not met. Therefore, another type of correlation has to be selected. The most obvious choice is the Spearman correlation because it relies on nearly all the same assumptions as the Pearson correlation, but it does not rely on normality, and the data can be ordinal as well; thus, it is a non-parametric test. Therefore, Spearman’s rank correlation was selected for the analysis.

Spearman’s coefficient (ρ) is used to measure the monotonic correlation between two variables [37]. A monotonic function is a function of one variable which is either entirely increasing or decreasing. The Spearman correlation coefficient is defined as the Pearson correlation coefficient between the rank variables [38].
$$\rho =1-\frac{6\sum d_{i}^{2}}{n\left(n^{2}-1\right)}$$
where *d* is the difference between the values of rank *x_i_* and rank *y_i_* and *n* is the number of observations.

Figure 2 summarizes the results of Spearman correlation analysis in a correlation matrix, where the values of the correlation coefficient are depicted by color and number in a heatmap.

This figure was created based on the filtered dataset. The correlation matrix of the entire dataset, as well as the correlation matrix of the difference between the two, can be found in the Appendix A. 

The results indicate that there is one very strong and one strong relationship. Product yield (Y) and product selectivity (S) have a perfect association, which can be expected, because these two parameters are derived from each other (Figure 2). Therefore, from the MIRA21 model’s point of view, it is possible to reduce the parameter set. Another strong correlation relationship was found between the quantity of the starting material (DNT) and the temperature (T) (Figure 2). Based on this, the higher the amount of starting material, the higher the reaction temperature used. Related to this, the correlation coefficient of the catalyst material quantity–temperature pair is 0.41. Accordingly, in the case of larger starting materials (DNT, CAT), the catalysts are tested at higher temperatures. 

A moderate relationship was identified between the active metal content (M) and CAT. These data are also derived from each other, so the relationship between them is clear. Thus, the higher correlation coefficient value of CAT-M-TON-T is not surprising. 

A promising finding was that maximum conversion and active metal content of the catalyst have a negative moderate association (ρ = −0.54). This would suggest that the conversion decreased as the amount of the active component of the catalyst increased. This question must be circumvented in order to draw a correct conclusion. A similar pattern of results was obtained in the case of the product yield–reaction time pair (ρ = −0.45). Temperature–pressure, metal content–pressure and temperature–selectivity relationships can also be characterized by moderate correlation coefficients between 0.4 and 0.42.

### 2.4. Assessment of DNT Hydrogenation Database

To understand the influence of reaction and catalytic parameters on catalyst performance, pair plot diagrams were used. Pair plot analysis sets include several scatter plots in one plot. To make it even more informative, each scatter plot diagram was colored according to different parameters. As a result, a large number of data were visualized for further analysis (Figure 3, Figure 4 and Figure 5). 

Each scatter plot uses points to represent the values of two different parameters. The location and dispersion of the points can be used to understand the nature of data. In addition to the representation of data pairs as points, a distribution function is also displayed (Figure 3). Different colors of the distribution function correspond to the type of catalyst support as shown in the pair plot. Only the distribution of the most common carriers is shown.

A pair plot diagram can confirm or deny the conclusions derived from the correlation matrix. The selectivity–yield plot clearly depicts the linear relationship between the data that is not even refuted by the separation according to the catalyst carriers. The already-mentioned high association between temperature and DNT molar amount during the correlation analysis is no longer so clear. In this case, even after data filtering, there is an outlier in the system, which can significantly affect the correlation coefficient.

Based on the available data, it is difficult to find trends related to the catalyst carriers. There are no particularly good groups that would indicate the best carriers. Therefore, in a specific type of carrier, both low and high performance were experienced. There are also transition metals, carbon, polymer, and zeolite catalysts which are effective. It is possible to identify transition metal–metal oxide combinations such as NiFe_2_O_4_ and CdFe_2_O_4_ that worked with particularly poor selectivity. Catalysts without carriers require more active components, but their catalytic activity performance varies significantly. In the analysis of the 2,4-dinitrotoluene hydrogenation reaction, the selectivity results provide the most information, as catalysts can be distinguished based on them. From the distribution point of view, TMO-supported catalysts need lower active metal content and could reach better selectivity under certain conditions. In the case of catalysts with a transition metal support, a classification can be made because of the analysis, which divides well-functioning and poorly functioning catalysts into separate groups. The data points of the carbon-based and zeolite-based catalysts are in one pile, providing a much more homogeneous result.

If we look at the distribution of the results according to the active component, a completely different picture can be drawn (Figure 4). Initially, the results were examined according to nickel, platinum, and palladium active components. Based on our experience and results, nickel was examined separately. Thus, to be able to examine the noble metal catalysts, nickel was omitted from the depicted pair plot analysis. Catalysts containing platinum form a separate group in several places. Platinum-type catalysts have less precious metal content and achieve maximum conversion in a shorter time. The performance of these catalysts is excellent. They have 100% conversion and a relatively high TON value.

Choosing the solvent to use for the reaction is an important step in catalyst testing [39]. Figure 5 compares the pair plot analysis results by methanol and ethanol solvents. The dielectric constant of methanol is higher than that of ethanol. On this basis, we should find that experiments carried out in methanol achieve better results than their ethanol counterparts. However, the results are not so clear. Of course, in addition to the solvent effect, other factors such as pressure and temperature also affect the reaction. Looking at the conversion column, 100% conversion can be achieved, regardless of the type of solvent, and there are many combinations of additional reaction parameters. If the selectivity or yield is examined, it can be seen that the results are already significantly separated. The results of several tests with methanol were worse than those with ethanol. Of course, this result can also be caused by the inhomogeneity of our data.

Our data are summarized in Figure 6 according to the correlation analysis and pair plot analysis. Active metal content and maximum conversion, reaction time and product yield, and selectivity and temperature reveal moderate associations. Consequently, a separate representation of these relationships has been selected and presented as a function of catalyst composition (Figure 6, colored points). The first half of the composition name represents the active component, and the second half represents the catalyst carrier.

Although the conversion and metal content were negatively correlated during the analysis, no clear correlation can be seen in the 2D diagram. On the other hand, it is clear that the metal content used in different catalysts can be classified according to the active component. Nickel-type catalysts have much higher metal content than their carbon-based counterparts, both with and without support.

The time–yield relationship also showed a negative correlation result. There are insufficient data to prove the real existence of the correlation. It is possible that over time, production will decrease because of the formation of by-products. The composition of the catalysts here has already distinguished the results. It can be concluded that, apart from a single exception, transition metal oxide catalysts containing platinum form a well-defined group and therefore provide the best performance in the shortest time. Transition metal oxide catalysts containing palladium also produce high yields over a longer period. Other data points show a very large standard deviation.

A moderate positive correlation between selectivity and temperature would also require additional data, as the result is not clear. Looking at the colors, clusters can also be identified, which is due to the fact that the temperature only takes 7–8 different values. At a given temperature, selectivity shows a large standard deviation. In this case, platinum-containing TMO catalysts are distinguished, although higher temperatures are used during the experiments than those of palladium-containing catalysts.

## 3. Materials and Methods

### 3.1. Catalyst Library

In our previous work, a catalyst library was created regarding the 2,4-dinitrotoluene catalytic hydrogenation reaction based on the MIRA21 model [20]. This database contains 58 catalysts and the corresponding multiparametric characterization (see Appendix A) [40,41,42,43,44,45,46,47,48,49,50,51,52,53,54]. All in all, five types of descriptors were considered (Table 1).

The data collection and processing were carried out according to our previous work [20]. The catalysts were characterized according to the MIRA21 model, the reactions tested were 295–393 K and 1–50 atm, and two cases (98 and 150 atm) were tested [20]. The maximum conversion time ranged from a few minutes to a day. The average reaction time for 100% conversion was 60 min, while the best catalyst reaction time was less than 40 min. The initial amount of dinitrotoluene was between 0.002 and 0.3 mol. The concentration of catalyst active metals also differed greatly from 5.13 × 10^−7^ mol to 0.034 mol. 

The primary screening of literature was based on the quality of the journal and the date of publication. The secondary screening of articles was based on the content. The data included in the database were normalized and then weighted, and finally, the catalysts were associated with a MIRA21 number. These quantified data formed the basis of the comparison without the ranking. Before starting the statistical analysis, the examined variables were selected from the catalyst library, and thus, data selection was carried out and three descriptor types were considered (Table 1, colored cells). Only quantifiable data were examined, and thus, the sustainability parameters were not analyzed. On the other hand, there are insufficient data for the variables related to the physical properties of catalyst. Therefore, the catalyst properties were also excluded from the analysis. This made it possible to examine the relationship between catalyst performance and reaction conditions. Furthermore, we investigated the effect of the catalyst composition and reaction solvent. The studied three descriptor types included 10 different (blue colored cells) parameters for correlation analysis (Table 2). In addition, our goal was to examine four more parameters (Table 1, gray colored cells): types and number of active metal components, catalyst carrier, and solvent. 

### 3.2. Exploratory Data Analysis

Before performing data analysis for statistical or prediction purposes, for example, using machine learning, it is necessary to understand the raw data we are working on [26]. Understanding and evaluating the quality of data is necessary to detect and treat atypical or incorrect points to avoid possible errors that may affect the results of the analysis. 

One way to carry out this pre-processing is through exploratory data analysis. EDA can be divided into two parts [55]: Univariate analysis (exploring characteristics of a single variable);Bi/multivariate analysis (comparative analysis of multiple variables; if we compare the correlation of two variables, it is called bivariate analysis).

A bivariate correlation analyzes whether and how two variables are linearly consistent, i.e., if one variable changes linearly as the other variable changes.

Correlation is a bivariate analysis that measures the strength of the association between two variables and the direction of the relationship [56]. In terms of the strength of the relationship, the value of the correlation coefficient varies between +1 and −1. A value of ±1 indicates a perfect degree of association between the two variables. As the correlation coefficient value becomes closer to 0, the relationship between the two variables will be weaker. The direction of the relationship is indicated by the sign of the coefficient; a + sign indicates a positive relationship, and a − sign indicates a negative relationship. Usually, in statistics, we measure four types of correlations: Pearson correlation, Kendall rank correlation, Spearman correlation, and point-biserial correlation.

### 3.3. Data Visualization 

Data visualization is a critical step in the data science process. To enhance understanding, hybrid plots which combine the strengths of different chart types and help to avoid losing valuable information are used.

To visualize the distribution of the dataset and summarize statistics, the combination of violin and strip plots was applied. A violin plot is a combination of a box plot and a kernel density plot. The strip plot represents an implementation of a scatterplot that exactly shows the inner structure of the distribution, its sample size, and the location of the individual observations. The correlation coefficients between different variables are visualized in a correlation heatmap, which is a graphical representation of a correlation matrix. 

To visualize the relationships between each variable, a pair plot was used [57]. It produces a matrix of relationships between each variable in the data for an instant examination. It can also be a great starting point for determining the types of regression analysis to use. The plot is supplemented with kernel density estimate (KDE) along the diagonal, which provides the distribution of the data. We have a categorical variable within our data frame, and it can be used to visually enhance the plots and see trends and distributions for each category by coloring.

The EDA was carried out in a Python programming environment by using NumPy, Pandas, Seaborn, and Matplotlib libraries [58,59,60,61].

## 4. Conclusions

Our first aim with the MIRA21 model was to characterize heterogeneous catalysts with objective numerical data from three types of descriptors to standardize catalyst characterization. In the present work, the introduced ranking model was mathematically validated by applying an exploratory data analysis procedure.

Literature data of 2,4-dinitrotoluene hydrogenation catalysts for toluene diamine synthesis were collected. Thereafter, a database from these data was created and filtered, and a general statistical analysis was performed. Afterward, a correlation matrix with 10 different parameters was created, and the parameters with strong correlations were selected. To confirm the correlation analysis, a pair plot analysis was performed to support the correlation results on the one hand and to draw conclusions on the composition of the catalyst on the other hand. Based on EDA, information was obtained on the hydrogenation of the nitro group of nitro compounds containing an alkyl side chain. It was found that the platinum-containing transition metal oxide catalysts provide excellent performance during the 2,4-dinitrotoluene hydrogenation reaction in a short time with low active metal content.

The present findings also confirm that the MIRA21 model can characterize, rank, and qualify catalysts. In the model’s parameter system, the correlation analysis indicated areas that need to be improved in one place, namely the use of selectivity and yield. As a result of the exploratory data analysis, it has become clear which parameters play a key role within the studied system. These are the reaction time, selectivity, type, and content of the active component.

Although the size of this dataset is limited, the applicability of the EDA process is well represented. Furthermore, it is suitable for providing the appropriate basis for machine learning by examining the results and parameter set of the database.

## Figures and Tables

**Figure 1 ijms-24-11461-f001:**
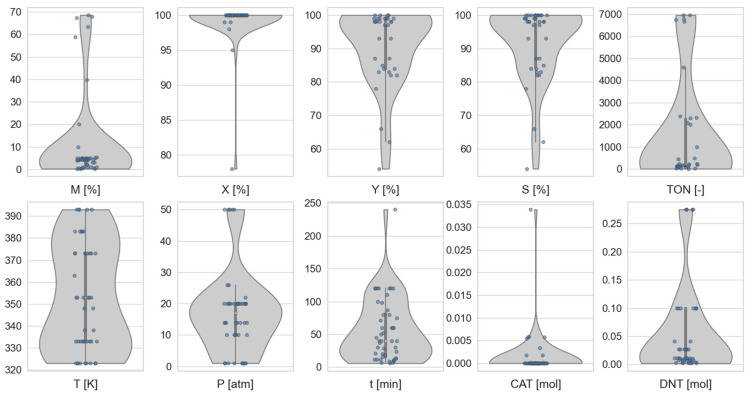
Violin plot diagram of the selected variables.

**Figure 2 ijms-24-11461-f002:**
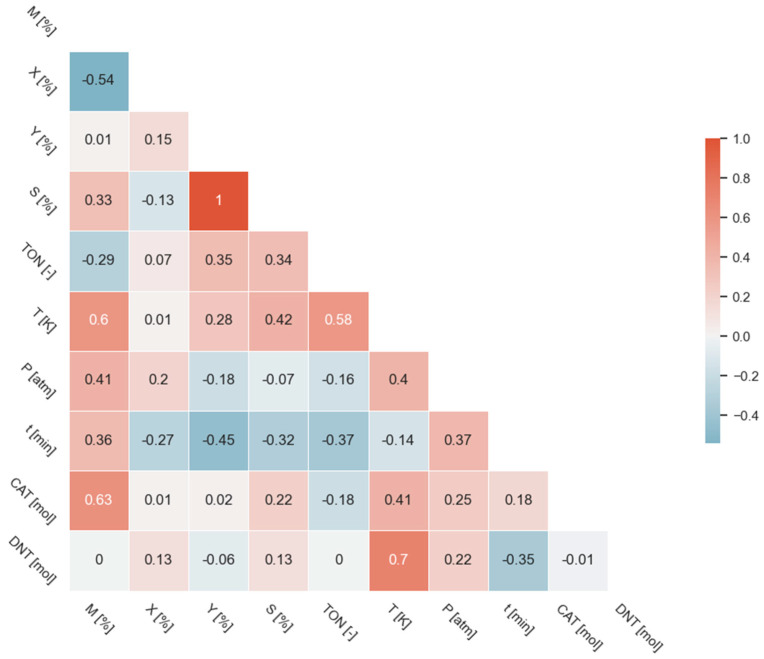
Results of Spearman correlation analysis.

**Figure 3 ijms-24-11461-f003:**
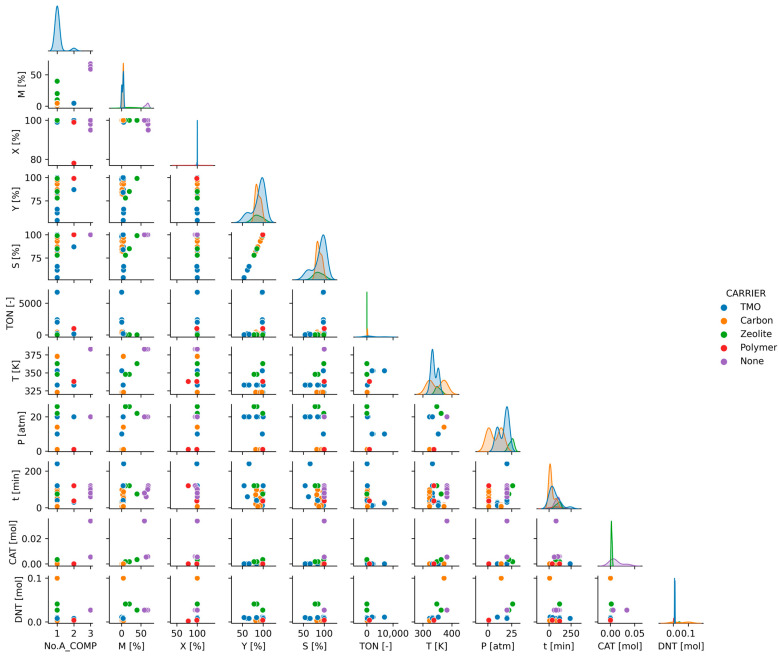
Pair plot analysis of descriptors by catalyst carrier (TMO—transition metal oxide).

**Figure 4 ijms-24-11461-f004:**
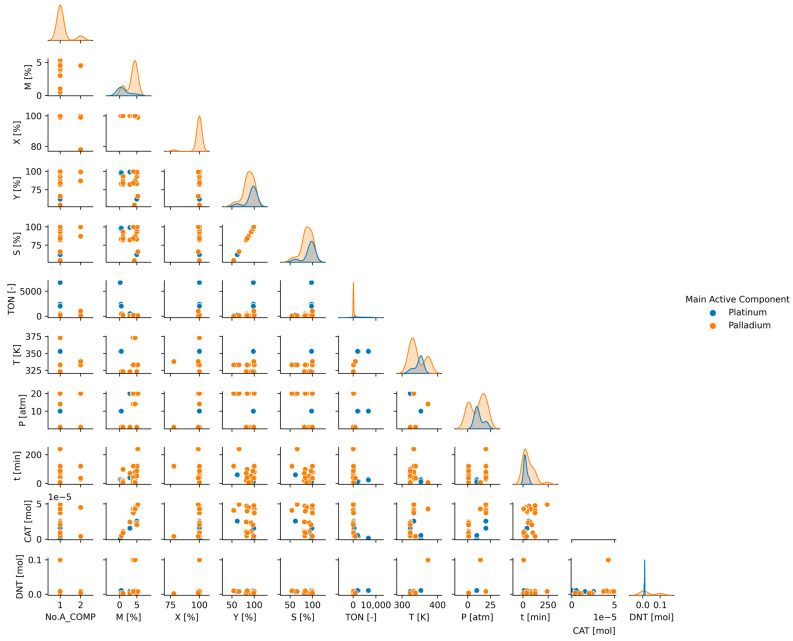
Pair plot analysis of descriptors by the catalyst’s main active component.

**Figure 5 ijms-24-11461-f005:**
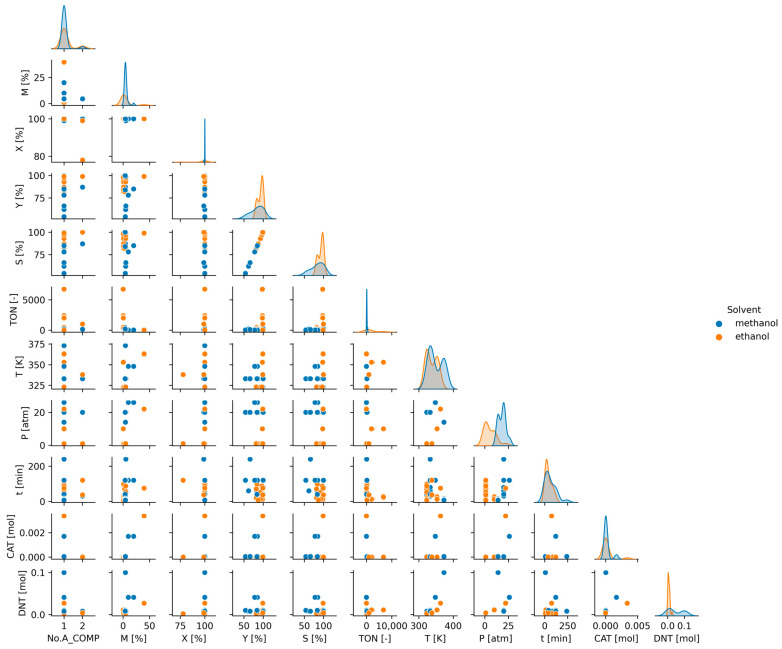
Pair plot analysis of descriptors by solvent.

**Figure 6 ijms-24-11461-f006:**
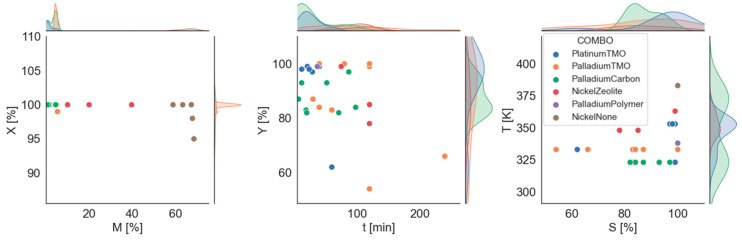
Scatterplot of active metal content–maximum conversion (**left**), time–yield (**middle**), and selectivity–temperature (**right**) by catalyst combinations (combination of catalyst compositions marked by different colors).

**Table 1 ijms-24-11461-t001:** Dataset of MIRA21. The cells marked in blue represent the parameters examined in the correlation analysis. The cells marked in gray indicate the parameters used for classification in the rest of the analysis.

Catalyst Composition	Catalyst Performance	Reaction Conditions	Catalyst Properties	Sustainability Parameters
Active metal content	Maximum conversion	Temperature	Catalyst particle size	Reactivation
Type of active metal content	Product yield	Pressure	Catalyst surface area	Stability
Number of active metals	Product selectivity	Time		Catalyst carrier
Catalyst carrier	Turnover number	Molar amount of catalyst		
		Molar amount of 2,4-dinitrotoluene		
		*Solvent*		

**Table 2 ijms-24-11461-t002:** Selected descriptors for statistical analysis.

	Notation	Name	Unit	Definition
	M	Catalyst composition	*w*/*w*%	Weight percentage of active component in the catalyst
Catalyst performance	X	Maximum conversion	mol%	Maximum desired product conversion achieved on a given catalyst
	Y	Product yield	mol%	Product yield for maximum conversion
	S	Product selectivity	mol%	Product selectivity for maximum conversion
	TON	Turnover number	-	Number of moles of product formed per 1 mol active metal when the maximum conversion reached
Reaction conditions	T	Temperature	K	Reaction temperature for maximum conversion
	P	Pressure	atm	Reaction pressure for maximum conversion
	t	Time	min	Time required to reach maximum conversion
	CAT	Molar amount of initial catalyst	mol	The molar amount of the active metal involved in the reaction—in the case of several metals, the sum of molar numbers
	DNT	Molar amount of starting substance	mol	The initial amount of starting substance involved in the reaction—2,4-dinitrotoluene

## Data Availability

Not applicable.

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
