# Peer review of "Towards Machine Learning in Heterogeneous Catalysis—A Case Study of 2,4-Dinitrotoluene Hydrogenation"

_ijms, 2023, doi:10.3390/ijms241411461_

Round 1

Reviewer 1 Report

The selection of parameters could be explained further, and the reaction system being studied could be elaborated on more, as well as the typical experimental measurements conducted in the collected literature. Some catalyst parameters such as carrier surface area, and some reactor parameters, such as hydrogen flow/bubble size, may be important to consider. Since yield and selectivity are so related, I am wondering why they were identified to be separate parameters in the first place. I have highlighted several errors in the submitted article, and made further comments that would improve the quality of the study. In the current form I cannot recommend publication.

Author Response

Review 1

Comments and Suggestions for Authors

The selection of parameters could be explained further, and the reaction system being studied could be elaborated on more, as well as the typical experimental measurements conducted in the collected literature. Some catalyst parameters such as carrier surface area, and some reactor parameters, such as hydrogen flow/bubble size, may be important to consider. Since yield and selectivity are so related, I am wondering why they were identified to be separate parameters in the first place. I have highlighted several errors in the submitted article, and made further comments that would improve the quality of the study. In the current form I cannot recommend publication.

We appreciate you for your precious time in reviewing our paper and providing valuable comments. The document has corrected editing and language errors and will answer questions and suggestions in this document.  

The selection of parameters could be explained further, and the reaction system being studied could be elaborated on more, as well as the typical experimental measurements conducted in the collected literature.

Thank you very much for your suggestion. We agree and make the following additions about the reaction and the parameters:

The catalysts were characterized according to the MIRA21 model, the reactions tested were 295-393 K, 1-50 atm, and two cases (98 and 150 atm) were tested. The maximum conversion time ranges from a few minutes to a day. The average reaction time for 100 % conversion was 60 minutes, while the best catalyst reaction time is less than 40 minutes. The initial amount of dinitrotoluene was between 0.002 and 0.3 mol. The concentration of catalyst active metals also differed greatly from 5.13*10-7 mol to 0.034 mol.

Some catalyst parameters such as carrier surface area, and some reactor parameters, such as hydrogen flow/bubble size, may be important to consider. Since yield and selectivity are so related, I am wondering why they were identified to be separate parameters in the first place.

Thank you for your suggestions. The choice of appropriate and sufficient parameters is a serious challenge. Our previous article about MIRA21 model describes the process and how we selected these parameters in detail. During this process, we try to optimize the number and quality of parameters most closely related to the data found in scientific publications.

In characterization, selectivity and yield appeared because research on the preparation of databases has shown that research results use one of the two parameters, alternating. Since it is certainly informative data, we do not want to leave it out of the system. We know that there is stronger or weaker correlation between certain parameters. Our goal is to discover the correlation between these parameters.

This database contains 58 catalysts and the corresponding multiparametric characterization.

Comment:

How about the nature of the reaction testing apparatus’ used, bubbling of H2, etc. as a parameter?

Thank you for your question.

The choice of appropriate and sufficient parameters is a serious challenge. Our previous article about MIRA21 model describes the process and how we selected these parameters in detail. During this process, we try to optimize the number and quality of parameters most closely related to the data found in scientific publications.

In characterization, selectivity and yield appeared because research on the preparation of databases has shown that research results use one of the two parameters, alternating. Since it is certainly informative data, we do not want to leave it out of the system. We know that there is stronger or weaker correlation between certain parameters. Our goal is to discover the correlation between these parameters.

Sustainability parameters

Comment:

 Consideration of source of active component, nature of active component (precious metal vs. abundant). What about the carrier is considered related to sustainability?

In the case of sustainability parameters, we refer to parameters specific to the reaction system. Because this parameter class does not include economic parameters, we do not focus on the active components of catalysts. On the other hand, the active components of the catalyst were considered for the design and prediction of the catalyst from data.

Since there is a relatively small amount of data available for this reaction, the results carry some uncertainty. However, the applicability of exploratory data analysis procedure to other reactions is undeniable.

Comment:

Can you comment on the amount of data recommended?

Previously, we have conducted studies on catalysts suitable for hydrogenation of nitrobenzene, testing 154 catalyst datasets, and preparing studies with more than 300 catalysts, i.e. around 12,000 data points. These databases are appropriate for drawing much more reliable and clearer conclusions.

Another strong correlation relationship was found between the quantity of the starting material (DNT) and the temperature (T) . Based on this, the higher the amount of starting material, the higher the reaction temperature used. Related to this, the correlation coefficient of the catalyst material quantity-temperature pair is 0.41.

Comment:

This seems intuitive, as it is an exothermic reaction so more reactant and more catalyst should correlate with reaction temperature.

Thank you for your comment. The temperature in the reactor will indeed depend on the maleness of both the catalyst and the reactant. For most of the research processed, the experimental conditions (thermostatting) were chosen so that the reaction could be conducted at a constant temperature. In agreement with the reviewer's suggestion, it is essential to take into account the thermochemistry and rate of the reaction when designing the reactor. However, this area is outside the scope of this article.

…it was clear that nickel is a separate class.

Comment: not a separate plot for this one?

Thank you for your comment. We corrected this sentence.

Based on the experience and results, nickel was examined separately.

On this basis, from the first approach,…

Comment: what approach was this?

Thank you for your question. I checked this sentence. The first approach expression means the previous sentence: The dielectric constant of methanol is higher than that of ethanol.

Because of the ambiguity, We removed this part of the sentence.

Thanks again for your comments and suggestions and we hope we have answered your questions satisfactorily.

Reviewer 2 Report

The article presents an example of data-driven analysis to compare activity of heterogeneous catalysts used for 2,4-dinitrotoluene hydrogenation. The presented tool can have potential applications in catalyst design. The article is well written and data presented in a clear and concise way. The conclusions regarding the parameters of the most active catalysts are not surprising (e.g., high activity of noble-metal catalysts), but the study demonstrates the potential of the data-driven analysis for the rational catalyst design. I recommend it for publication after some issues are resolved (see below).

Authors state that the selection of the data was based on the quality of journal (among others). What criteria was adopted to define it?

Based on Violin plots some of the data was excluded. How large is the data set left after this step? Is it the dataset present in Supplementary Information?

Analysis of the results indicate that data regarding zeolite-based and carbon-based materials are in one pile (page 9). Can it be related to usually high specific surface area of these systems? Authors are encouraged to look into this parameter of the catalysts. 

The following editorial-like issues should also be fixed:

1. Table 1 has coloured fields but the meaning of different colours is not clearly explained - the legend should be added (in text, green and grey are mentioned, in Table blue and grey is seen).

2. There are references (to tables/ figures) missing in the text.

3. The reference to the article introducing MIRA21 (reference 20) does not use correct abbreviation of the journal name.

4. Please correct "platina" to "platinum" (lines 333-334)

Author Response

Reviewer 2

Comments and Suggestions for Authors

The article presents an example of data-driven analysis to compare activity of heterogeneous catalysts used for 2,4-dinitrotoluene hydrogenation. The presented tool can have potential applications in catalyst design. The article is well written and data presented in a clear and concise way. The conclusions regarding the parameters of the most active catalysts are not surprising (e.g., high activity of noble-metal catalysts), but the study demonstrates the potential of the data-driven analysis for the rational catalyst design. I recommend it for publication after some issues are resolved (see below).

Authors state that the selection of the data was based on the quality of journal (among others). What criteria was adopted to define it?

Thank you for your question. In the analysis, we selected scientific publications from journals ranked Q1 or Q2. This was a primary selection criterion to increase the reliability of the data.

Based on Violin plots some of the data was excluded. How large is the data set left after this step? Is it the dataset present in Supplementary Information?

Thank you for your comment. This observation is correct. The data were filtered and then continued the study with 48 catalysts. This filtered database can be found in the Supplementary Information. In order to avoid misunderstandings, we have corrected the title of the first Annex:

“Filtered dataset of statistical analysis according to MIRA21”

Analysis of the results indicate that data regarding zeolite-based and carbon-based materials are in one pile (page 9). Can it be related to usually high specific surface area of these systems? Authors are encouraged to look into this parameter of the catalysts. 

We thank the reviewer for pointing this out. The phenomenon is due to the above-mentioned large specific surface, but the database does not provide an opportunity to show it. This information cannot be extracted only from the database, as in most cases there is a lack of specific surface area data. However, a much larger database is being prepared for a similar reaction where we will examine this relationship.

The following editorial-like issues should also be fixed:

  1. Table 1 has coloured fields but the meaning of different colours is not clearly explained - the legend should be added (in text, green and grey are mentioned, in Table blue and grey is seen)

Thank you for this comment. We have fixed the error.

  1. There are references (to tables/ figures) missing in the text.

Thank you for this comment. We checked all the references in the text.

  1. The reference to the article introducing MIRA21 (reference 20) does not use correct abbreviation of the journal name.

Thank you for this comment. We have updated the name to “MDPI Catalysts”.

  1. Please correct "platina" to "platinum" (lines 333-334)

Thank you for your comment. I corrected it.

Thanks again for your comments and suggestions and we hope we have answered your questions satisfactorily.

Reviewer 3 Report

In this study, the authors analyzed 2,4-dinitrotoluene hydrogenation through the MIRA21 model descriptor system. The manuscript is well written and concise, and my recommendation is to accept the manuscript for publication after the authors consider the following comment: perhaps the authors should additionally emphasize the experimental applicability of their method, i.e. correlate it with the experimental results of the hydrogenation of 2,4-dintrotoluene, as well as other catalytic processes. To what extent the optimized conditions are really experimentally applicable?

Author Response

Review 3

Comments and Suggestions for Authors

In this study, the authors analyzed 2,4-dinitrotoluene hydrogenation through the MIRA21 model descriptor system. The manuscript is well written and concise, and my recommendation is to accept the manuscript for publication after the authors consider the following comment: perhaps the authors should additionally emphasize the experimental applicability of their method, i.e. correlate it with the experimental results of the hydrogenation of 2,4-dintrotoluene, as well as other catalytic processes. To what extent the optimized conditions are really experimentally applicable?

We appreciate you for your precious time in reviewing our paper and providing valuable comments.

The purpose of our data analysis is to provide a basis for the design of both experimental designs and catalysts from these published experimental results and the information extracted from them. The information used for experimental design in scientific and industrial environments is the result of catalyst conditions or catalyst composition.

The catalyst rankings obtained based on the MIRA21 model are also suitable for the systematic design of catalysts as they highlight physical and/or chemical properties that should be considered as part of the design strategy. In the process of material development, for example, in the case of a specific carrier, the model and data analysis results of the MIRA21 are used to receive advice and suggestions for the decoration of the carrier to enhance efficiency.

The optimal experimental conditions depend on the catalyst. It cannot be generally determined but can be specified based on the catalyst database.

Thanks again for your comments and suggestions and we hope we have answered your questions satisfactorily.
